# Hydraulic Integrated Interconnected Regenerative Suspension: Sensitivity Analysis and Parameter Optimization

Sijing Guo [1], Liang Chen [1], Yu Pan [2], Xuxiang Wang [1] and Gangfeng Tan [1,*]

[1] School of Automotive Engineering, Wuhan University of Technology, Wuhan 430070, China
[2] Department of Mechanical Engineering, Virginia Tech, Blacksburg, VA 24060, USA
[*] Correspondence: auto_nova@whut.edu.cn

**Abstract:** Hydraulic integrated interconnected regenerative suspension (HIIRS) is a novel suspension system that can simultaneously harvest the vibration energy in the suspension and enhance the vehicle dynamics. The parameter sensitivity of the HIIRS system is analyzed and the significant parameters are optimized in this paper. Specifically, a half-vehicle model with the HIIRS is established. Based on the model, the parameter sensitivity of the hydraulic system is analyzed with three objectives, ride comfort, road holding, and average energy harvesting power. The parameters considered in this study are more abundant than those in previous related studies, including hydraulic cylinder inner diameter, hydraulic motor displacement, resistance, initial system pressure, and accumulator parameters. It turns out that the most sensitive parameters are the inner diameter of the hydraulic cylinder, the resistance, and the displacement of the hydraulic motor. To further study the performances that the HIIRS could present, both the single-objective optimization and the multi-objective optimization problems are solved and compared with the optimized traditional suspensions. The optimized HIIRS performs better in ride comfort and road holding than the optimized traditional suspension and anti-roll bar suspension. Different from the previous suspension optimization design, multi-objective optimization not only considers the traditional performance of the suspension but also incorporates the energy harvesting characteristics into the optimization objective. In the multi-objective optimization, a Pareto front is obtained, which shows that the ride comfort conflicts with the road holding and the energy harvesting power, while road holding and energy harvesting power did not conflict. The Pareto front shows that the optimized HIIRS is superior to the traditional suspension in ride comfort and road holding, and also harvests considerable energy.

**Keywords:** hydraulically interconnected suspension; energy harvesting suspension; sensitivity analysis; optimization

## 1. Introduction

Rollovers of vehicles have always been an extremely dangerous type of traffic accident. In the United States, a total of 6291 people died in rollover accidents in 2019, accounting for 28.3% of the annual traffic accident deaths [1]. The suspension system, as one of the important components in automobiles, has a significant impact on the roll movement of the vehicle, as well as the ride comfort and the road holding [2]. While performing the functions, a considerable amount of vibration energy is dissipated in the suspension. It was reported that around 10% of the fuel energy was consumed in the suspension when driven on the road [3]. By harvesting this part of the energy, the fuel efficiency could be improved by 5% and 6%, respectively, for heavy vehicles and off-road vehicles [4]. To alleviate the rollover-induced accidents and harvest the vibration energy, this paper studies a novel suspension system, the hydraulic integrated interconnected regenerative suspension (HIIRS), which achieves the structural features and advantages of the hydraulic interconnected suspension (HIS) and the energy harvesting suspension.

By interconnecting the suspension in various manners, such as the mechanical interconnection [5], the pneumatic interconnection [6,7], and the hydraulic interconnection [8], the interconnected suspension could achieve superior dynamics performances. Mechanically interconnected suspensions, such as anti-roll bars, are widely used for their simple structure and high reliability. Citroen vehicles equipped with it demonstrated greater roll resistance but deteriorated ride comfort [5]. The air-interconnected suspension realized superior ride comfort, but it presented low bearing capacity and required strong sealing and large space [9]. The hydraulic interconnected suspension (HIS) could significantly improve the anti-roll ability of the vehicle without affecting the bounce stiffness, which was considered a promising suspension system [8]. Zhang et al. [10] proposed a frequency-domain model of HIS based on the hydraulic impedance method and verified the accuracy of the model through bench tests. Cao et al. proposed a time-domain model [11] and evaluated the effect of the piston area and the accumulator parameters on the dynamics performances [12]. Ding et al. [13] proved the anti-pitch function of the HIS on a three-axle truck while maintaining the ride comfort at the same level.

The energy harvesting suspensions can be divided into two types, the mechanical type [14,15] and the hydraulic type [16,17]. Compared with the mechanical energy harvester, the hydraulic one is more reliable under heavy-duty working conditions [18]. Guo et al. [17] developed a prototype of an electromagnetic shock absorber for heavy vehicles. The experimental results showed that an average power of 220 W could be harvested under a harmonic excitation of 3 Hz-7 mm. Zhang et al. [19] studied a half-bridge hydraulic energy-harvesting shock absorber, which harvested 33.4 W at the excitation of 1.67 Hz and 50 mm.

The hydraulic interconnected energy harvesting suspension, which combined the features of the energy harvesting suspension and the HIS, has become increasingly popular in the past five years. Wang et al. [20] proposed a hydraulic interconnected energy regenerative suspension with two motor-generator assemblies and studied the multi-mode control system and its optimal design. Zou et al. [21] proposed a hydraulic interconnected suspension-based energy-harvesting shock absorber (HIS-HESA) and showed that the HIS-HESA harvested an average electrical power of 190 W. Guo [22] studied a hydraulic integrated interconnected regenerative suspension system (HIIRS). Simulation results showed that the average energy harvesting power of the HIIRS reached 186 W when the off-road vehicle was driven at the speed of 36 km/h on a Class C road. Qin et al. [23] proposed a new energy-harvesting hydraulic interconnected suspension (EH-HIS) and demonstrated 60% and 11% enhancements over the traditional suspension, respectively, under the emergency steering and the braking condition. The simulation results showed that the EH-HIS harvested an average power of 215 W when the corresponding SUV was driven at a speed of 60 km/h on a Class D road.

For the newly proposed suspensions, studying the parameter sensitivity and optimization is considered a significant way to achieve an optimal suspension design. Shen et al. [24] used the structure-immittance approach to optimize the inerter-spring-damper suspension (ISD) and showed that the optimized suspension reduced the root mean square of the vehicle body vertical acceleration and the pitch acceleration by 31% and 35%, respectively. Zhou et al. [25] used the Sobol's method to study the parameter sensitivity in the hydraulic interconnected suspension (HIS) and showed that the valve connected to the hydraulic cylinder had a greater impact on the bounce response and the valve connected to the accumulator had a greater impact on the roll response, which provided a theoretical basis for the further optimization design. Zhou et al. [26] proposed a Human-knowledge-integrated Particle Swarm Optimization (Hi-PSO) scheme to globally optimize the design of the hydraulic-electromagnetic energy-harvesting shock absorber (HESA) for road vehicles. An average energy efficiency of 59.07% was achieved in the test duty cycles. Li et al. [27] studied the parameter sensitivities and the multi-objective optimization problem for an energy harvesting suspension. Results showed that ride comfort and handling stability were two contradictory performance indicators.

In previous studies, sensitivity analysis and parameter optimization have played a huge role in suspension studies to pursue favorable performances. However, few studies about parameter analysis and optimization have been conducted for the newly proposed HIIRS. Therefore, this paper holistically analyzes and optimizes the performances of the HIIRS by considering the ride comfort, the road holding, and the energy harvesting power. A half-vehicle system equipped with the HIIRS is modeled, and its acceleration responses, dynamic tire load, and energy harvesting power on random roads are calculated and treated as the objective functions. The parameter sensitivity to the objectives is studied and the most sensitive parameters are focused on the optimization problems. Both the single-objective optimization and multi-objective optimization are then performed and compared with the optimized traditional suspensions in the paper.

The rest of the paper is organized as follows. Section 2 builds the model of a half-vehicle equipped with the HIIRS system; Section 3 calculates the performance indices; Section 4 introduces the sensitivity analysis method and sorts the sensitivity of each parameter; Section 5 optimizes the HIIRS by considering the ride comfort, the road holding, the energy harvesting power and also the trade-off among the three indices. Section 6 summarizes the conclusions. The description of terms in the manuscript is shown in Table A1 in Appendix A.

## 2. Modeling

In this section, the working principle of the HIIRS is introduced first. Then, to compare the performances among the HIIRS, the traditional spring-damper suspension, and the traditional suspension with an anti-roll bar, the corresponding half-vehicle models are built.

### 2.1. Working Principle of the HIIRS

The structure and working principle of HIIRS are shown in Figure 1. It includes two hydraulic cylinders, two hydraulic rectifiers composed of check valves, a hydraulic motor-generator unit, a high-pressure accumulator, and a low-pressure accumulator.

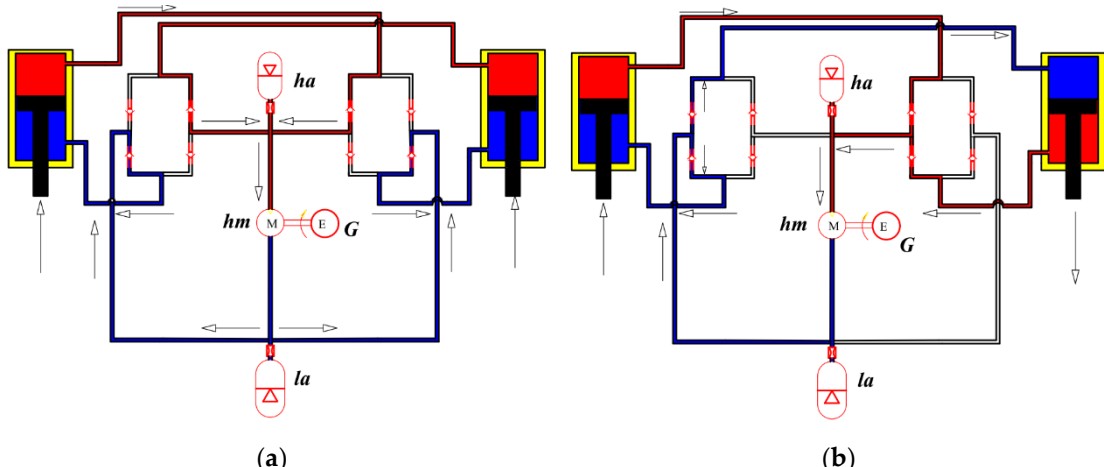

**(a)**　　　　　　　　　　　　　　　　　　　**(b)**

**Figure 1.** The structure and working process of the HIIRS in the (**a**) bounce motion and (**b**) roll motion (*la*—low-pressure accumulator; *ha*—high-pressure accumulator; *hm*—hydraulic motor; *G*—generator).

The hydraulic rectifier uses four check valves to convert the bidirectional fluid flow out of the hydraulic cylinder to the unidirectional flow when it passes the hydraulic motor. When the wheels are excited in the same direction, the vehicle body bounces. As shown in Figure 1a, the high-pressure oil from the hydraulic cylinders flows through two rectifier valve blocks and converges to the inlet of the hydraulic motor. The high-pressure accumulator here can stabilize the fluid flow through the hydraulic motor and ensure a stable rotary speed of the hydraulic motor and the generator for high efficiency. There is

a low-pressure accumulator at the outlet of the hydraulic motor, which can compensate for the change in oil volume in the HIIRS system. The low-pressure oil returns to the low-pressure chamber of the hydraulic cylinder through two rectifier valve blocks. The pressure difference between the two chambers of the hydraulic cylinders on both sides produces vertical force in the same direction to resist body bouncing. When the wheels are excited in the opposite direction, the body rolls. As shown in Figure 1b, the high-pressure oil from the high-pressure chamber flows through the same rectifier valve block and converges to the inlet of the hydraulic motor. The low-pressure oil at the outlet of the hydraulic motor returns to the low-pressure chamber of the hydraulic cylinder through another rectifier valve block. The pressure difference between the chambers of the hydraulic cylinders on both sides produces a vertical force in different directions, which exerts an anti-roll torque on the vehicle body. In such a way, the HIIRS holds the potential to harvest the vibration energy into electricity and achieve an anti-bounce ability and an anti-roll ability simultaneously.

### 2.2. Modeling of the HIIRS System

The model built in this study is a four-degree-of-freedom, roll-plane half-vehicle equipped with a hydraulic integrated interconnected regenerative suspension (HIIRS) system. The mechanical subsystem in the model is shown in Figure 2. The hydraulic subsystem is shown in Figure 3. The body and the tires are connected by springs and double-acting hydraulic cylinders. The parameters of the mechanical subsystem are shown in Table 1. The parameters of a SUV are selected since the HIIRS could harvest the considerable vibration energy of the SUV when it was driven on a bumpy road and also enhance the anti-roll ability for a SUV with a relatively high center of gravity.

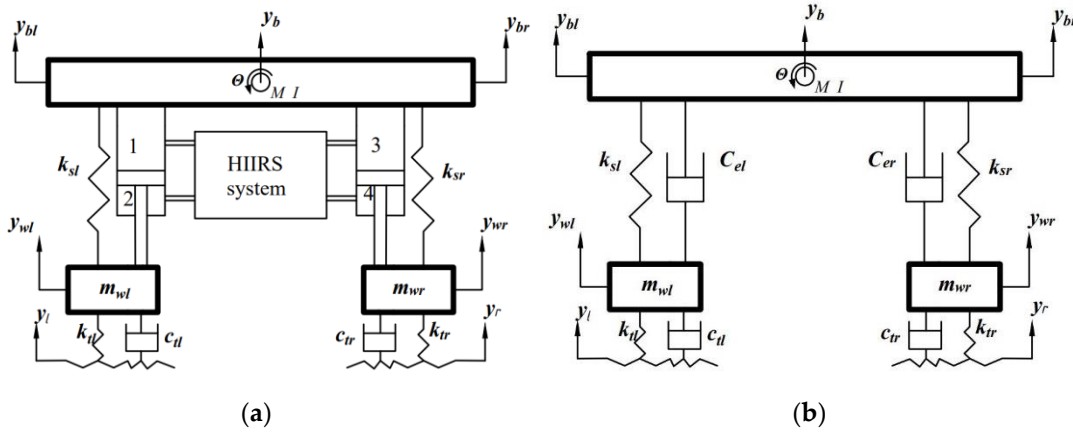

**(a)**                                                                 **(b)**

**Figure 2.** Schematic of a half-vehicle with (**a**) the HIIRS system, and (**b**) the traditional suspension.

**Table 1.** The SUV parameters used in the mechanical subsystem.

| Symbol | Value | Units | Description |
|--------|-------|-------|-------------|
| $M$ | 1400 | kg | Sprung mass |
| $m_j$ | 105 | kg | Unsprung mass ($j = l, r$ = left, right) |
| $I$ | 523 | Kgm$^2$ | Sprung mass moment of inertia about the roll axis |
| $b_j$ | 0.825 | M | Distance from c.g. to suspension strut ($j = l, r$ = left, right) |
| $k_{sj}$ | 112 | kN/m | Mechanical suspension spring stiffness |
| $k_{tj}$ | 1200 | kN/m | Tire stiffness |
| $c_{tj}$ | 300 | Ns/m | Tire damping coefficient |
| $C_e$ | 3800 | Ns/m | Traditional suspension damping coefficient |

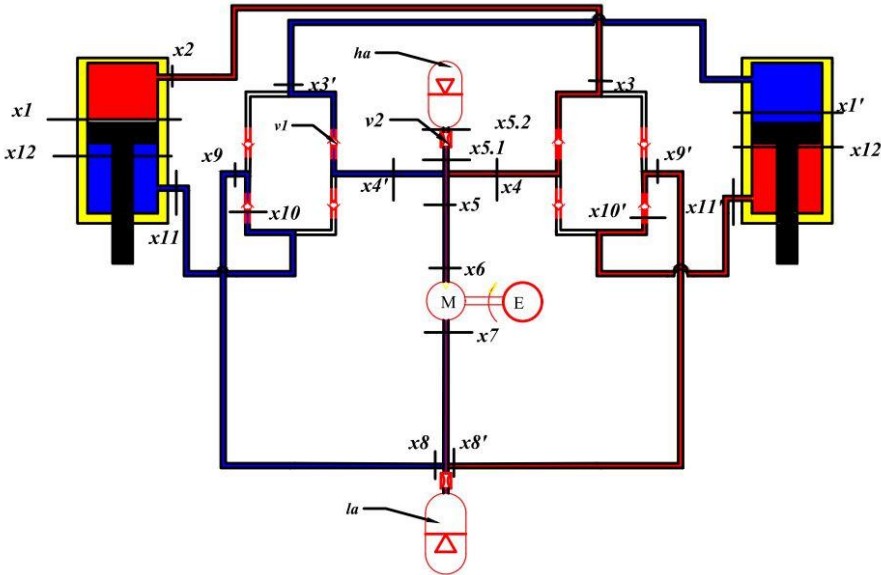

**Figure 3.** Schematic of the HIIRS system (*v1*—Check valves; *la*—low-pressure accumulator; *ha*—high-pressure accumulator; *v2*—accumulator valve; *x*, *x′*—state nodes).

According to the boundary conditions of the hydraulic-mechanical system, the relationship between the flow rate vector *q* of the fluid system and the displacement *y* of the mechanical system is

$$q(t) = AD_2\dot{y} \tag{1}$$

where $D_2 = \begin{bmatrix} 1 & 0 & -1 & b_l \\ 1 & 0 & -1 & b_l \\ 0 & 1 & -1 & -b_r \\ 0 & 1 & -1 & -b_r \end{bmatrix}$, $q(t) = [q_1, q_2, q_3, q_4]^T$, $A = diag([A_1, A_2, A_3, A_4])$, and

$y = [y_{wl}, y_{wr}, y_b, \theta]^T$. $q_i$ and $A_i$, respectively, represent the flow and cross-sectional area of the *i*-th chamber of the hydraulic cylinders, and the chamber numbers are shown in Figure 2. $y_{wl}, y_{wr}, y_b$ and $\theta$ are the left wheel displacement, right wheel displacement, body displacement, and body roll angle of the half-vehicle model, respectively.

The linear relationship exists between the flow vector $Q(s)$ and pressure vector $P(s)$ of the hydraulic system in the frequency domain.

$$Q(s) = Z^{-1}(s)P(s) \tag{2}$$

where *Z* is the impedance matrix, which is obtained by multiplying the impedance matrix of each element in the hydraulic system.

The relationship between the upper and lower hydraulic chambers is

$$\begin{bmatrix} P_2 \\ Q_2 \end{bmatrix} = T^a \begin{bmatrix} P_1 \\ Q_1 \end{bmatrix} \begin{bmatrix} P_4 \\ Q_4 \end{bmatrix} = T^b \begin{bmatrix} P_3 \\ Q_3 \end{bmatrix} \tag{3}$$

where $T^a$ and $T^b$ are the impedance matrices of the left and right hydraulic circuits, respectively; $P_i$ and $Q_i$ are the flow and pressure of the *i*-th chamber of the hydraulic cylinders, respectively (in the frequency domain).

According to Equations (2) and (3), $Z(s)$ can be written as

$$Z(s) = \begin{bmatrix} -\frac{T^a_{11}}{T^a_{12}} & \frac{1}{T^a_{12}} & 0 & 0 \\ T^a_{21} - \frac{T^a_{22}T^a_{11}}{T^a_{12}} & \frac{T^a_{22}}{T^a_{12}} & 0 & 0 \\ 0 & 0 & -\frac{T^b_{11}}{T^b_{12}} & \frac{1}{T^b_{12}} \\ 0 & 0 & T^b_{21} - \frac{T^b_{22}T^b_{11}}{T^b_{12}} & \frac{T^b_{22}}{T^b_{12}} \end{bmatrix} \tag{4}$$

matrices $T^a$ and $T^b$ are obtained by multiplying the impedance matrices of the components in the hydraulic system, that is

$$T^{a,b} = \prod_{i=1}^{i=11} T_{i,i+1} \tag{5}$$

where $T_{i,i+1}$ represents the impedance matrix between node $i$ and node $i+1$, the elements of the matrix are shown in Table 2 and the distribution of nodes is shown in Figure 3.

**Table 2.** The elements of impedance matrix.

| Impedance Matrix | Description | Elements |
|---|---|---|
| $T_{1,2}$ | Impedance matrix of upper chamber of hydraulic cylinder | $T_{1,2} = \begin{bmatrix} cosh\Gamma(s) & -Z_C(s)sinh\Gamma(s) \\ -\frac{sinh\Gamma(s)}{Z_C(s)} & cosh\Gamma(s) \end{bmatrix}$ |
| $T_{2,3}$ | Impedance matrix of hydraulic pipeline | $T_{2,3} = \begin{bmatrix} cosh\Gamma(s) & -Z_C(s)sinh\Gamma(s) \\ -\frac{sinh\Gamma(s)}{Z_C(s)} & cosh\Gamma(s) \end{bmatrix}$ |
| $T_{3,4}$ | Impedance matrix of check valve | $T_{3,4} = \begin{bmatrix} 1 & -R_{v1} \\ 0 & 1 \end{bmatrix}$ |
| $T_{4,5}$ | Impedance matrix of high-pressure accumulator unit | $T_{4,5} = \begin{bmatrix} 1 & 0 \\ \frac{1}{Z_A-R_{V2}} & 1 \end{bmatrix}$, $Z_A = -\frac{\gamma \bar{P}^2}{s p_p v_p}$ |
| $T_{5,6}$ | Identity matrix | $T_{5,6} = \begin{bmatrix} 1 & 0 \\ 0 & 1 \end{bmatrix}$ |
| $T_{6,7}$ | Impedance matrix of energy harvesting unit | $T_{6,7} = \begin{bmatrix} 1 & -Z_M \\ 0 & \eta_v \end{bmatrix}$, $Z_M = \frac{4\pi^2 k_e k_t \eta_v}{(R_e+R_{in})q_m^2 \eta_m} + \frac{4J_m \pi^2 \eta_v}{q_m^2 \eta_m}s$ |
| $T_{7,8}$ | Identity matrix | $T_{7,8} = \begin{bmatrix} 1 & 0 \\ 0 & 1 \end{bmatrix}$ |
| $T_{8,9}$ | Impedance matrix of low-pressure accumulator unit | $T_{8,9} = \begin{bmatrix} 1 & 0 \\ \frac{1}{Z_A-R_{V2}} & 1 \end{bmatrix}$ |
| $T_{9,10}$ | Impedance matrix of check valve | $T_{9,10} = \begin{bmatrix} 1 & -R_{v1} \\ 0 & 1 \end{bmatrix}$ |
| $T_{10,11}$ | Impedance matrix of hydraulic pipeline | $T_{10,11} = \begin{bmatrix} cosh\Gamma(s) & -Z_C(s)sinh\Gamma(s) \\ -\frac{sinh\Gamma(s)}{Z_C(s)} & cosh\Gamma(s) \end{bmatrix}$ |
| $T_{11,12}$ | Impedance matrix of upper chamber of hydraulic cylinder | $T_{11,12} = \begin{bmatrix} cosh\Gamma(s) & -Z_C(s)sinh\Gamma(s) \\ -\frac{sinh\Gamma(s)}{Z_C(s)} & cosh\Gamma(s) \end{bmatrix}$ |

### 2.3. The Half-Vehicle Models with Various Suspensions

To compare the performances among the HIIRS, the conventional spring-damper suspension, and the anti-roll bar suspension, the corresponding half-vehicle models are built in this section.

### 2.3.1. The Half-Vehicle Model with the HIIRS

According to Newton's second law, the kinematics equation of the half-car model is written by

$$M\ddot{y} + C\dot{y} + Ky = f(t) \tag{6}$$

where the displacement vector $y = [y_{wl}, y_{wr}, y_b, \theta]$, $M = \begin{bmatrix} m_l & 0 & 0 & 0 \\ 0 & m_r & 0 & 0 \\ 0 & 0 & M & 0 \\ 0 & 0 & 0 & I \end{bmatrix}$, $C =$

$\begin{bmatrix} c_{tl} & 0 & 0 & 0 \\ 0 & c_{tr} & 0 & 0 \\ 0 & 0 & 0 & 0 \\ 0 & 0 & 0 & 0 \end{bmatrix}$, $K = \begin{bmatrix} k_{sl} + k_{tl} & 0 & -k_{sl} & b_l k_{sl} \\ 0 & k_{sr} + k_{sl} & -k_{sr} & -b_r k_{sr} \\ -k_{sl} & -k_{sr} & k_{sl} + k_{sr} & -b_l k_{sl} + b_r k_{sr} \\ b_l k_{sl} & -b_r k_{sr} & -b_l k_{sl} + b_r k_{sr} & b_l^2 k_{sl} + b_r^2 k_{sr} \end{bmatrix}$. $f(t)$ can be

divided into two parts: one is the force from the hydraulic cylinder, and the other is from the road roughness. If the vector $p = [p_1, p_2, p_3, p_4]^T$ represents the pressure of the four chambers of the hydraulic cylinder, the matrix $A = diag([A_1, A_2, A_3, A_4])$ represents the cross-sectional area of the four chambers of the hydraulic cylinder, and $\xi = [\xi_l, \xi_r]^T$ represents the excitation of the road surface. Then Equation (6) can be rewritten as

$$M\ddot{y} + C\dot{y} + Ky = D_1 A p(t) + \vec{F}\xi \tag{7}$$

where linear transformation matrix $D_1 = \begin{bmatrix} -1 & 1 & 0 & 0 \\ 0 & 0 & -1 & 1 \\ 1 & -1 & 1 & -1 \\ -b_l & b_l & b_r & -b_r \end{bmatrix}$.

By substituting Equation (2) into the Laplace form of Equation (7), the frequency domain equation of the system can be obtained as

$$[s^2 M + s\overline{C}(s) + K]Y(s) = F_x(s) \tag{8}$$

where $\overline{C}(s) = C - D_1 A Z(s) A D_2$, $F_x(s) = \vec{F}(s)\vec{\xi}(s)$ is the force exerted by the road on the tire, and $\vec{\xi}(s) = [\xi_l, \xi_r, 0, 0]^T$ is the road excitation. $\vec{F}$ is a $4 \times 4$ matrix whose elements are zero except for the first two diagonals. The first two diagonal elements of $\vec{F}$ are $\vec{F}_{11}(s) = k_{tl} + sc_{tl}$ and $\vec{F}_{22}(s) = k_{tr} + sc_{tr}$.

Assuming $B(s) = s^2 M + s\overline{C}(s) + K$, Equation (8) can be rewritten as

$$B(s)Y(s) = \overline{F}(s)\overline{\xi}(s) \tag{9}$$

The frequency response matrix of the half-car system is defined as

$$H_y(s) = \frac{Y(s)}{\overline{\xi}(s)} = B^{-1}(s)\overline{F}(s) \tag{10}$$

In Equation (10), the frequency response matrix describes the system displacement response to any excitations. Hence, by giving a known excitation, the responses of the HIIRS system can be completely determined and the vibration analysis can be carried out in the same way as other linear systems.

### 2.3.2. The Half-Vehicle Model with a Conventional Spring-Damper Suspension

With Newton's second law, the equation of a traditional half-vehicle is given by

$$M\ddot{y} + C_{ts}\dot{y} + Ky = \vec{F}\xi \tag{11}$$

where $C_{ts} = \begin{bmatrix} c_{tl} + c_{el} & 0 & -c_{el} & b_l c_{el} \\ 0 & c_{er} + c_{tr} & -c_{er} & -b_r c_{er} \\ -c_{el} & -c_{er} & c_{el} + c_{er} & 0 \\ b_l c_{el} & -b_r c_{er} & 0 & b_l^2 c_{el} + b_r^2 c_{er} \end{bmatrix}$. Matrices $M$ and $K$ are consistent with those in Equation (6).

By conducting Laplace transform, Equation (11) can be written as

$$\left(Ms^2 + Cs + K\right)Y(s) = \overline{F}(s)\overline{\overline{\xi}}(s) \tag{12}$$

The transfer matrix of a traditional suspension can be written as

$$H_{Ts}(s) = \frac{Y(s)}{\overline{\overline{\xi}}(s)} = \left(Ms^2 + Cs + K\right)^{-1}\overline{F}(s) \tag{13}$$

2.3.3. The Half-Vehicle Model with an Anti-Roll Bar Suspension

For the half-vehicle model installed with an anti-roll bar (ARB) suspension, the difference from the traditional suspension is that the ARB suspension adds a matrix $K_a$ to the original stiffness matrix $K$, as

$$H_{At}(s) = \frac{Y(s)}{\overline{\overline{\xi}}(s)} = \left(Ms^2 + C_{ts}s + K + K_a\right)^{-1}\overline{F}(s) \tag{14}$$

where $K_a = \begin{bmatrix} \frac{k_a}{l_a^2} & -\frac{k_a}{l_a^2} & 0 & -\frac{k_a}{l_a} \\ -\frac{k_a}{l_a^2} & \frac{k_a}{l_a^2} & 0 & \frac{k_a}{l_a} \\ 0 & 0 & 0 & 0 \\ -\frac{k_a}{l_a} & \frac{k_a}{l_a} & 0 & k_a \end{bmatrix}$, $k_a$ is the stiffness of the anti-roll bar and $l_a$ is the distance between the anti-roll bar mounting points.

## 3. Performance Indices

In this section, the random road excitation of a half-vehicle was introduced. Then, to compare the performances between the HIIRS and the traditional suspensions, the performance indices of the anti-roll ability, the ride comfort, and the road holding are determined. As a unique advantage of the HIIRS, the average energy harvesting power of the HIIRS when driven on a random road is also calculated.

### 3.1. Random Road Excitation

A road profile can be represented by its power spectral density (PSD) function. Assuming that the road surface is a two-dimensional (2D) Gaussian uniform and isotropic random process [28], the self-spectral density of the left tracks, the right tracks, and the cross-spectrum between the left and right tracks are equal, (i.e., $S_{ll}(\omega) = S_{rr}(\omega) = S_D(\omega)$).

The spectral density function of one-sided road excitation can be written as

$$S_q(n) = cn^{-2w} \tag{15}$$

where $w$ is a coefficient ranging from 1 to 1.25, and is generally taken as 1.

Considering the coherence between the left and right wheels, the road input spectral density matrix of the half-vehicle model is

$$S = \begin{bmatrix} S_D & S_X \\ S_X & S_D \end{bmatrix} \tag{16}$$

where $S_D$ is the self-power spectral density of the road excitation $S_D = \frac{1}{u}S_q(n) = \frac{1}{u}cn^{-2w}$, and $S_X$ is the cross-power spectral density $S_X = \frac{1}{u}[2c\left(\frac{\pi L}{n}\right)^w / \Gamma(w)]J_w(2\pi Ln)$ [29]. In the equation of the $S_X$, $L$ is the wheel track between the left wheel and the right wheel as 1.65, $J_w$ is the second-class modified Bessel function of order $w$, and $\Gamma(w)$ is the gamma function, $\Gamma(w) = \int_0^\infty e^{-w}t^{w-1}dt$. $u$ is the vehicle speed, which is taken as 36 km/h in this study. $c$ is the road roughness coefficient, which is selected as $c = 256 \times 10^{-8}$ in this study, representing the Class C road.

### 3.2. Ride Comfort

The relationship between the power spectral density of the response $Y$ and the power spectral density of the excitation $X$ is

$$Y = |H|^2_{Y \sim X} X \tag{17}$$

where $H_{Y \sim X}$ is transfer function from $X$ to $Y$.

For the half-vehicle system in this study, it is a multiple-input and multiple-output system. The power spectrum of its outputs can be expressed as

$$S_i(f) = [H^*_{i1} \ H^*_{i2}] S \begin{bmatrix} H_{i1} \\ H_{i2} \end{bmatrix} \tag{18}$$

where $S_i(f)$ represents the power spectrum of the i-th output, $H_{i1}$ and $H_{i2}$, respectively, represent the transfer function of the *i*-th output to the first and second inputs, and $*$ represents the conjugate complex number.

Therefore, the power spectral density function of the vehicle body bounce acceleration and the roll angular acceleration of the half-vehicle model can be written as

$$
\begin{aligned}
G_{accb}(f) &= [H^*(3,1) H^*(3,2)] S \begin{bmatrix} H(3,1) \\ H(3,2) \end{bmatrix} \\
G_{accr}(f) &= [H^*(4,1) H^*(4,2)] S \begin{bmatrix} H(4,1) \\ H(4,2) \end{bmatrix}
\end{aligned} \tag{19}
$$

where $G_{accb}(f)$ is the power spectral density function of the bounce acceleration, and $G_{accr}(f)$ is the power spectral density function of the roll acceleration.

In this study, the root-mean-square (RMS) values of the performances are used as evaluation functions, since the vibration responses present the same probability of positive and negative. The RMS values can be calculated by

$$\sigma = \sqrt{\int G(f) df} \tag{20}$$

According to ISO2631-15 [30], the ride comfort of a vehicle can be judged by the weighted acceleration which combines the vehicle body bounce acceleration and the roll acceleration. Hence, the performance index of the ride comfort $J_1$ is written as

$$J_1 = \sqrt{k_b^2 \int w_k^2(f) G_{accb}(f) df + k_r^2 \int w_e^2(f) G_{accr}(f) df} \tag{21}$$

where $k_b$ and $k_r$ are the weighting coefficients for the bounce acceleration and the roll acceleration and their values are 1 and 0.63, respectively [30]. $w_k(f)$ and $w_e(f)$ are the frequency weighting functions of bounce acceleration and roll acceleration, respectively, as

$$
w_k(f) = \begin{cases} 0.5 & (0.5 < f < 2) \\ f/4 & (2 < f < 4) \\ 1 & (4 < f < 12.5) \\ 12.5/f & (12.5 < f < 80) \end{cases}, \ w_e(f) = \begin{cases} 1 & (0.5 < f < 1) \\ 1/f & (1 < f < 80) \end{cases}.
$$

### 3.3. Road Holding

The road holding ability is judged by the dynamic tire load, which is expressed as

$$F_t(s) = \overline{F}(s)\left(Y(s) - \overline{\zeta}(s)\right) \tag{22}$$

Combining Equations (10) and (22), the transfer function of the dynamic tire load can be written as

$$H_{F_t}(s) = F_t(s)/\overline{\zeta}(s) = \overline{F}(s)\left(B^{-1}(s)\overline{F}(s) - I\right) \tag{23}$$

Hence, the evaluation function $J_2$ of road-holding can be calculated by

$$J_2 = \sqrt{\int \left[ H_{F_t}^* \ H_{F_t}^* \right] S \begin{bmatrix} H_{F_t} \\ H_{F_t} \end{bmatrix} df} \tag{24}$$

$J_2$ represents the root mean square of the dynamic load of the tire. Excessive dynamic load of the tire is detrimental to the stability of the vehicle in a straight line.

### 3.4. Energy-Harvesting Power

In addition to the vibration response, another key indicator of the HIIRS system is the vibration energy-harvesting power. The HIIRS system uses an energy-harvesting unit composed of a hydraulic motor and an electric motor to convert the vibration energy.

The energy-harvesting power is closely related to the parameters of the energy-harvesting unit and the hydraulic system, and can be calculated by

$$\begin{cases} P_h = I^2 R_e \\ I = \frac{U_{emf}}{R_e + R_{in}} \\ U_{emf} = k_e \omega \\ \omega = 2\pi \frac{Q_M}{q_m} \eta_v \end{cases} \tag{25}$$

where $P_h$ is the energy-harvesting power, $R_e$ is the external resistance, $I$ is the current, $U_{emf}$ is the induced electromotive force, $R_{in}$ is the internal resistance of the generator, $k_e$ is the speed constant of the generator, $Q_M$ is the inlet flow rate of the hydraulic motor, $q_m$ is the motor displacement, and $\eta_v$ is the volumetric efficiency.

In Equation (25), only $Q_M$ is the unknown quantity. The power spectral of $Q_M$ is obtained by

$$G_{Q_M} = H_{Q_M Y}^{*T} G_Y H_{Q_M Y} \tag{26}$$

where $G_Y$ is the power spectral density matrix $G_Y = diag\left( \left[ G_{y_{wl}}, G_{y_{wr}}, G_{y_b}, G_\theta \right] \right)$, $H_{Q_M Y}$ is the transfer matrix between the motor inlet flow rate and the vibration response $y = [y_{wl}, y_{wr}, y_b, \theta]$. $H_{Q_M Y}^{*T}$ is the transposed conjugate matrix of $H_{Q_M Y}$.

Combining Equations (25) and (26), the objective function $J_3$ of power output can be written as

$$J_3 = \frac{4\pi^2 \eta_v^2 k_e^2 R_e}{(R_e + R_{in})^2 q_m^2} \overline{Q_M} \tag{27}$$

where $\overline{Q_M}$ is the average flow rate at the hydraulic motor inlet.

## 4. Sensitivity Analysis

In this section, one of the sensitivity analysis methods, the Morris method, is introduced, and the parameter sensitivity of the HIIRS to the objective functions in Section 3 is analyzed.

### 4.1. Morris Method

The Morris screening method is suitable for models with a large number of uncertain inputs or high computational costs [31]. The core of this method is to construct the sample space to generate the random experimental geometry of "one variable change at a time". Take out a set of independent parameters $X = [x_1, x_2, \ldots x_m]$, map them to the interval [0, 1], take out one of the variables $x_i$, change it by a fixed amount $\Delta$, and form a new set of parameters $X_i$. Such changes become a "trajectory". The sensitivity index of the parameter $x_i$ corresponding to each trajectory is

$$E_{x_i} = \frac{f(x_1, \ldots, x_{i-1}, x_i + \Delta, x_{i+1}, \ldots, x_m) - f(X)}{\Delta} \tag{28}$$

Sort the parameters according to the average value of the calculated sensitivity index. The parameter with the highest average value of the sensitivity index is considered to be the most significant parameter, while the lowest parameter is relatively insignificant. The range and description of the parameters to be analyzed are shown in Table 3.

**Table 3.** Parameter description and the corresponding design space in the Morris method.

| Symbol | Range | Units | Description |
|---|---|---|---|
| $P_{ph}$ | $[1 \times 10^5, 3 \times 10^6]$ | Pa | Pre-charge pressure of high-pressure accumulator |
| $V_{ph}$ | $[1 \times 10^{-4}, 1 \times 10^{-3}]$ | m³ | Pre-charge gas volume of high-pressure accumulator |
| $P_{pl}$ | $[1 \times 10^5, 3 \times 10^6]$ | Pa | Pre-charge pressure of low-pressure accumulator |
| $V_{pl}$ | $[1 \times 10^{-4}, 1 \times 10^{-3}]$ | m³ | Pre-charge gas volume of low-pressure accumulator |
| $R$ | $[0.5, 300]$ | Ω | Circuit external resistance |
| $D_c$ | $[10, 70]$ | mm | Inner diameter of hydraulic cylinder |
| $q_m$ | $[5 \times 10^{-6}, 1 \times 10^{-4}]$ | m³/rev | Hydraulic motor displacement |
| $D_p$ | $[10, 30]$ | mm | Inner diameter of hydraulic pipeline |
| $P$ | $[1 \times 10^5, 3 \times 10^6]$ | Pa | Initial pressure of hydraulic system |

### 4.2. Sensitivity Analysis Result

The obtained sensitivity indices of the seven parameters are shown in Table 4 and Figure 4. Figure 4 shows the sensitivity factors of each parameter and the cumulative sensitivity ratio. It can be seen that the three most sensitive parameters are the hydraulic cylinder's inner diameter $D_c$, the displacement of the hydraulic motor $q_m$ and the electric resistance $R$. Among the three parameters, the most significant parameter for $J_1$ and $J_2$ is $D_c$, the second was $q_m$, and the third was $R$. For the objective function $J_3$, the most significant parameter was $R$, the second was $D_c$, and the third was $q_m$. In addition, the cumulative sensitivity of the three parameters $D_c$, $q_m$ and $R$ accounted for more than 90% of the objectives. Hence, only these three parameters were selected in the optimizations.

**Table 4.** Sensitivity of the parameters.

| Parameter | $D_c$ | $q_m$ | $R$ | $D_p$ | $P$ | $P_{ph}$ | $V_{ph}$ | $P_{pl}$ | $V_{pl}$ |
|---|---|---|---|---|---|---|---|---|---|
| $J_1$ Sensitivity | 0.7907 | 0.0544 | 0.0444 | 0.0335 | 0.0124 | 0.0206 | 0.014 | 0.014 | 0.016 |
| $J_2$ Sensitivity | 0.8462 | 0.0343 | 0.0293 | 0.0263 | 0.0124 | 0.0165 | 0.0107 | 0.0126 | 0.0115 |
| $J_3$ Sensitivity | 0.1466 | 0.1405 | 0.6325 | 0.0054 | 0.0153 | 0.0146 | 0.0116 | 0.0179 | 0.0156 |

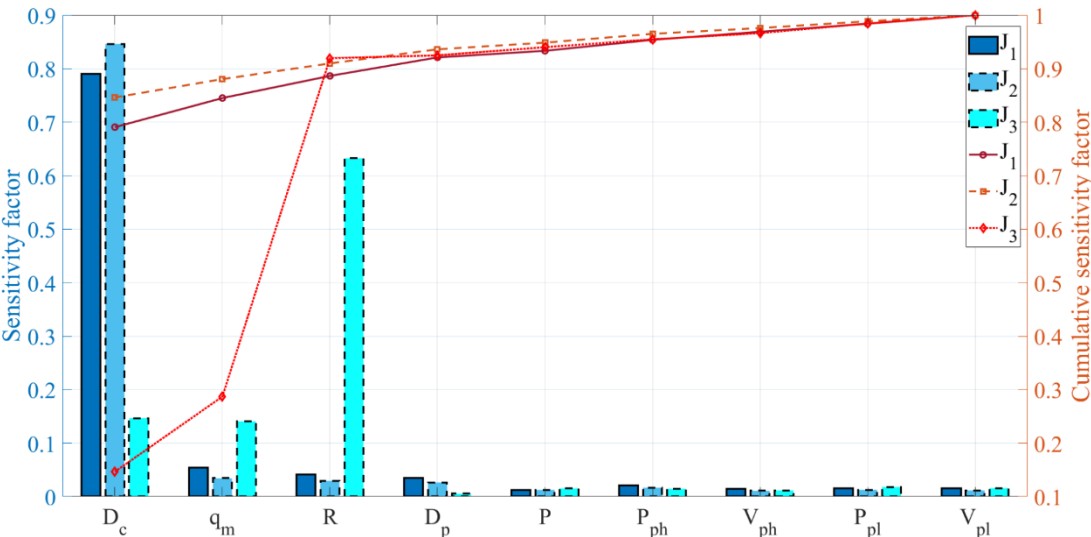

**Figure 4.** The sensitivity factors and the cumulative sensitivity factors of various parameters for the ride comfort $J_1$, the road holding $J_2$, and the energy-harvesting power $J_3$.

The objective functions $J_1$ and $J_2$ reflect the ride comfort and road holding of the vehicle, which are closely related to the damping of the vehicle. Therefore, the key parameters $D_c$, $q_m$ and $R$ of the damping matrix are naturally also the key parameters of $J_1$ and $J_2$.

In the three key parameters, the inner diameter of the hydraulic cylinder $D_c$ is positively correlated with the damping, according to Equation (29). The damping of a suspension could be calculated by

$$\overline{C_s} = F_s/v_s = (\Delta p \frac{\pi}{4} D_c^2 + p_2 \frac{\pi}{4} D_p^2)/v_s \tag{29}$$

where $F_s$ and $v_s$ are the output force of the hydraulic cylinder and the moving speed of the piston rod, respectively. $\Delta p = p_1 - p_2$, $p_1$ represents the pressure of the non-rod chamber of the hydraulic cylinder, and $p_2$ represents the pressure of the rod chamber. Equation (29) shows that damping $\overline{C_s}$ has a quadratic relationship with $D_c$. Meanwhile, larger $D_c$ results in a larger flow rate out of the hydraulic system, and thus a larger hydraulic resistance to be overcome and inducing a larger pressure difference $\Delta p$. In such a way, $\Delta p$ was positively correlated with $D_c$. Hence, there is a positive correlation between the damping and the hydraulic cylinder's inner diameter $D_c$.

The displacement of the hydraulic motor $q_m$ is negatively correlated with the damping. The impedance matrix $T_{6,7}$ in Table 2 shows the relationship between $q_m$ and the impedance $Z_m$ of the hydraulic motor, as $Z_M = \frac{4\pi^2 k_e k_t \eta_v}{(R+R_{in})q_m^2 \eta_m} + \frac{4J_m \pi^2 \eta_v}{q_m^2 \eta_m}s$. It can be seen that $Z_M$ is negatively correlated with $q_m$, and $Z_M$ is part of the impedance matrix $Z(s)$ of the system. Therefore, $q_m$ is negatively correlated with the damping. Although $q_m$ is one of the key parameters influencing the damping, its impact is not as significant as $D_c$, since $q_m$ could only affect $\Delta p$ in Equation (29) while $D_c$ is positively correlated with both $\overline{C_s}$ and $\Delta p$.

The resistance $R$ is also negatively correlated with the damping, which has been confirmed in other studies [32]. In the frequency domain model of the HIIRS, similar to $q_m$, the resistance $R$ is also negatively correlated with $Z_M$, but the impact of $R$ is not as significant as $q_m$. This is reflected in the impedance formula of the motor, as $Z_M = \frac{4\pi^2 k_e k_t \eta_v}{(R+R_{in})q_m^2 \eta_m} + \frac{4J_m \pi^2 \eta_v}{q_m^2 \eta_m}s$. Therefore, the resistance $R$ is negatively correlated with the damping, and its impact was less than $q_m$ and $D_c$.

As for the objective function $J_3$ representing the energy-harvesting power, the influences of various parameters on it could be reflected in Equation (27), which could be written as

$$J_3 = \frac{4\pi^2 \eta_v^2 k_e^2 R}{(R + R_{in})^2 q_m^2}\overline{Q_M} = k_P \overline{Q_M} \tag{30}$$

where $k_P = 4\pi^2 \eta_v^2 k_e^2 \frac{1}{q_m^2(R+\frac{R_{in}^2}{R}+2R_{in})}$, $\overline{Q_M}$ is the average flow rate at the hydraulic motor inlet.

The effect of $D_c$ on the energy-harvesting power is reflected through $\overline{Q_M}$. $\overline{Q_M}$ is mainly affected by two factors, the cylinder diameter $D_c$ and the moving speed $v_s$ of the piston rod. $D_c$ is positively correlated with the damping which hinders the movement of the piston rod. That is, $D_c$ is negatively correlated with $v_s$, while $D_c$ is positively correlated with $\overline{Q_M}$. Thus, it is difficult to determine whether $D_c$ is positively or negatively correlated with the energy-harvesting power. For this reason, the effect of $D_c$ on the energy-harvesting power is proved to be not the most significant.

As for the displacement of the hydraulic motor $q_m$, it is negatively correlated with the damping which would hinder the movement of the piston rod and induce a smaller $v_s$ and further a smaller $\overline{Q_M}$. Hence, $q_m$ is positively correlated with $\overline{Q_M}$, while $q_m$ is negatively correlated with $k_P$. Such contradiction makes $q_m$ not the most significant parameter for the energy-harvesting power.

In general, the resistance $R$ is negatively correlated to the energy-harvesting power. Resistance $R$ is negatively correlated with the damping which hinders the movement of the piston rod $v_s$. However, Equation (30) shows that when the resistance $R$ is greater than the internal resistance $R_{in}$, $k_P$ is negatively correlated with $R$. In this study, the internal

resistance $R_{in} = 0.28\ \Omega$, and the variation range of the resistance $R$ in the optimization is [0.5, 300]. Thus, the resistance $R$ is negatively correlated with $k_P$. Among the key parameters, the influence of resistance $R$ on road holding and ride comfort is the least significant, therefore, the positive effect of resistance $R$ on damping and $\overline{Q_M}$ is not as significant as the negative effect of resistance $R$ on $k_P$.

## 5. Parameter Optimization

To enhance the performances of the HIIRS, the most sensitive parameters obtained in the last section are optimized in this section. Both single-objective optimization and multi-objective optimization are conducted in this section.

### 5.1. Single Objective Optimization

The genetic algorithm (GA) is used in single objective optimization for its fast convergence and great robustness [33].

### 5.1.1. Optimization of Ride Comfort

The ride comfort of the HIIRS, the traditional suspension (TS), and the suspension with an anti-roll bar (ARB) are optimized via minimize the objective function $J_1$. In the HIIRS, the three key parameters obtained in Section 4.2, namely the inner diameter of the hydraulic cylinder $D_c$, the resistance $R$, and the displacement of the hydraulic motor $q_m$, are optimized. In the traditional suspension, the damping coefficient $C_e$ is optimized. In the suspension with an anti-roll bar, the damping coefficient $C_e$ and the anti-roll stiffness $k_a$ are optimized. The parameter values and acceleration root mean square values before and after optimization are shown in Table 5.

**Table 5.** Ride comfort optimization solution.

| Parameters | Variation Range | Initial | Optimization Solution |
|---|---|---|---|
| Resistance $R$ | [0.5, 300] | 10 | 203 |
| Inner diameter of hydraulic cylinder $D_c$ | $\left[1 \times 10^{-2}, 7 \times 10^{-2}\right]$ | $5 \times 10^{-2}$ | $3.6 \times 10^{-2}$ |
| Displacement of hydraulic motor $q_m$ | $\left[5 \times 10^{-6}, 1 \times 10^{-4}\right]$ | $2 \times 10^{-5}$ | $7.3 \times 10^{-5}$ |
| Damping coefficient $c_e$ in TS | [1000, 10,000] | 3800 | 2438 |
| Anti$-$roll bar stiffness $k_a$ in ARB | $\left[1 \times 10^4, 5 \times 10^4\right]$ | $3.5 \times 10^4$ | $4.7441 \times 10^4$ |
| Damping coefficient $c_e$ in ARB | [1000, 10,000] | 3800 | 2282.3 |
| Bounce acceleration RMS of HIIRS | | 1.5155 | 0.7338 |
| Roll acceleration RMS of HIIRS | | 0.5762 | 0.4807 |
| Total weighted acceleration RMS of HIIRS | | 1.6628 | 0.7938 |
| Bounce acceleration RMS of TS | | 1.6205 | 0.9567 |
| Roll acceleration RMS of TS | | 0.5880 | 0.5855 |
| Total weighted acceleration RMS of TS | | 1.6866 | 1.0254 |
| Bounce acceleration RMS of ARB | | 1.6205 | 0.9541 |
| Roll acceleration RMS of ARB | | 0.5060 | 0.3416 |
| Total weighted acceleration RMS of ARB | | 1.6515 | 0.9781 |

Figure 5 shows that after optimization, the damping of TS and ARB is reduced and the ride comfort is improved. We can find a rule that smaller damping can bring better comfort to traditional suspension and suspension with ARB. As for the HIIRS, after optimization, the resistance $R$ of the HIIRS and the displacement $q_m$ of the motor are increased, and the inner diameter $D_c$ of the hydraulic cylinder is reduced, which illustrates that the damping of optimized HIIRS is smaller than that of the initial according to the analysis results in Section 4.2.

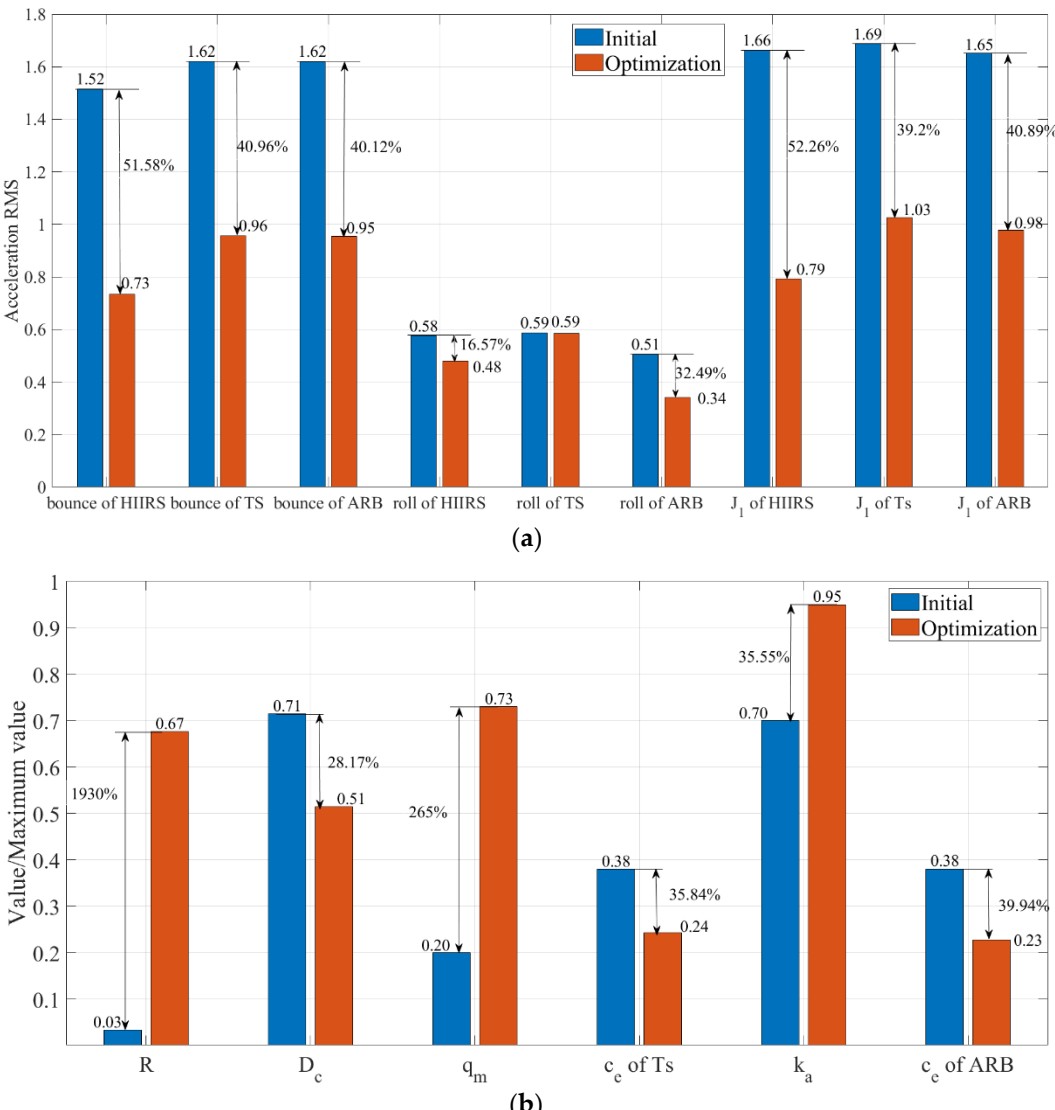

**Figure 5.** Performances and parameters comparison (**a**) performance comparison; (**b**) parameters comparison.

In order to more intuitively display the comfort of the three optimized suspensions, the bounce acceleration and roll acceleration power spectral density (PSD) curves of the three optimized suspensions were shown in Figure 6. Figure 6a shows the bounce acceleration PSD curve of optimized HIIRS, optimized traditional suspension (optimized TS), and optimized anti-roll bar (optimized ARB). Figure 6b shows the roll acceleration PSD curves of three optimized suspensions.

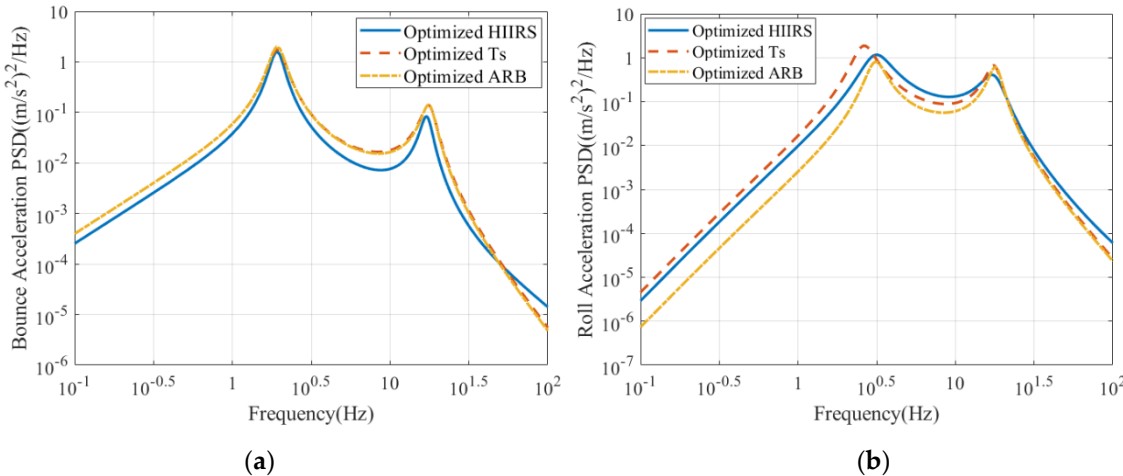

**Figure 6.** The optimization results with the ride comfort as the single objective, (**a**) the body bounce acceleration PSD; (**b**) the body roll acceleration PSD.

Figure 6b shows that the roll acceleration of the optimized ARB is always minimal. From this point of view, the anti-roll ability of ARB seems to be better than HIIRS. However, this may be because the objective of the optimization is ride comfort, and its evaluation index is the weighted acceleration of bounce acceleration and roll acceleration. Actually, we also took the roll acceleration as the optimization goal and conducted some research. By optimizing the HIIRS and the ARB, the RMS roll acceleration of the HIIRS and the ARB are, respectively, 0.168 rad/s$^2$ and 0.1705 rad/s$^2$, the anti-roll ability of the optimized HIIRS could be 1.47% higher than the ARB suspension.

### 5.1.2. Optimization of Road Holding

As shown in Section 4.2, the key parameters, the resistance $R$, the inner diameter of the hydraulic cylinder $D_c$, and the displacement of the hydraulic motor $q_m$ are selected for optimizing the road holding. The optimization results were shown in Table 6.

**Table 6.** Road holding optimization solution.

| Parameters | Variation Range | Initial | Optimization Solution |
|---|---|---|---|
| Resistance $R$ | $[0.5, \, 300]$ | 10 | 162 |
| Inner diameter of hydraulic cylinder $D_c$ | $\left[1 \times 10^{-2}, 7 \times 10^{-2}\right]$ | $5 \times 10^{-2}$ | $4.3 \times 10^{-2}$ |
| Displacement of hydraulic motor $q_m$ | $\left[5 \times 10^{-6}, 1 \times 10^{-4}\right]$ | $2 \times 10^{-5}$ | $9.2 \times 10^{-6}$ |
| Damping coefficient $c_e$ of Ts | $[1000, \, 10{,}000]$ | 3800 | 8541 |
| Damping coefficient $c_e$ of ARB | $[1000, \, 10{,}000]$ | 3800 | 9309.3 |
| Anti$-$roll bar stiffness $k_a$ | $\left[1 \times 10^4, 5 \times 10^4\right]$ | $3.5 \times 10^4$ | $7.5898 \times 10^4$ |
| Tire dynamic load RMS of HIIRS | | 2369.1 | 2172.2 |
| Tire dynamic load RMS of TS | | 3150.9 | 2696.9 |
| Tire dynamic load RMS of ARB | | 3110.5 | 2646.1 |

Table 6 shows results of the optimization results of the road holding for the three suspensions. The damping coefficients of traditional suspension and suspension with ARB increased from 3800 Ns/m to 8541 Ns/m and 9309.3 Ns/m, respectively, and the corresponding $J_2$ decreases by 14.42% and 14.93%, respectively. It could be inferred that a larger damping is beneficial to reduce the dynamic load of the tire. We also compare the ride comfort and road holding of HIIRS optimized for ride comfort and HIIRS optimized for road holding, the results are shown in Table 7. Compared with the HIIRS optimized for ride comfort, the HIIRS optimized for road holding has smaller resistance $R$, larger inner diameter $D_c$ and larger displacement $q_m$, which illustrates that the damping of the latter is larger than that of the former according to Section 4.2.

**Table 7.** Comparison of two optimized HIIRS.

| Optimization Objectives | Ride Comfort | Road Holding |
|---|---|---|
| Resistance $R$ | 203 | 162 |
| Inner diameter of hydraulic cylinder $D_c$ | $3.6 \times 10^{-2}$ | $4.3 \times 10^{-2}$ |
| Displacement of hydraulic motor $q_m$ | $7.3 \times 10^{-5}$ | $9.2 \times 10^{-6}$ |
| Ride comfort objective function $J_1$ | 0.7938 | 1.2236 |
| Road holding objective function $J_2$ | 2935.3 | 2172.2 |

Figure 7 shows that the tire dynamic load PSD curve of optimized HIIRS has three peaks, one more peak than the other two suspensions, and the additional peak frequency is 3.6 Hz. The tire dynamic load of the optimized HIIRS near this peak frequency is larger than that of the other two suspensions. This difference can be due to the "interconnection" characteristic of HIIRS. Since the left and right hydraulic cylinders are connected to the energy harvesting unit, as long as the hydraulic cylinder at one side is displaced, oil would pass through the energy harvesting unit and generate a pressure difference. Even if the ground at the other side is not excited, the hydraulic cylinder at that side will still move. Equations (10), (13), and (14) show the transfer matrix of HIIRS, traditional suspension, and suspension with an anti-roll bar. Based on this, the amplitude-frequency characteristics of the left wheel displacement to the right ground excitation could be drawn, as shown in Figure 8. It could be found that the curve of HIIRS also has an additional peak near 3.6 Hz. This shows that the ground excitation on the right side near 3.6 Hz will cause a large displacement of the left tire, and then produce a large tire dynamic load.

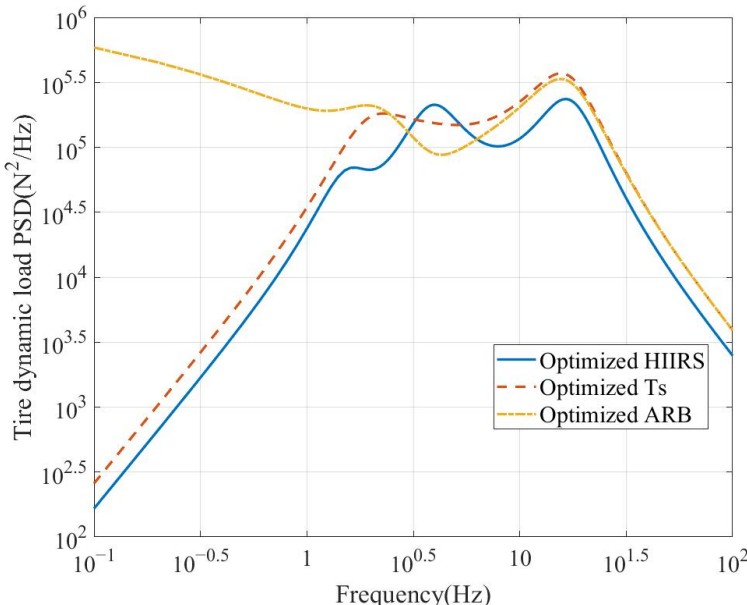

**Figure 7.** Road holding optimization effect.

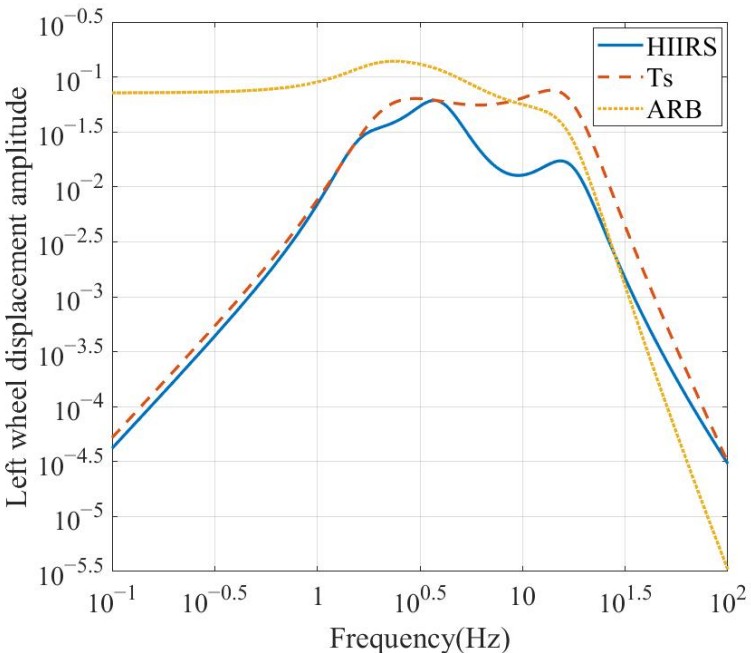

**Figure 8.** Amplitude frequency characteristics of left wheel displacement on right ground input.

To sum up, the genetic algorithm can effectively optimize the ride comfort and road holding of HIIRS. Compared with the optimized traditional suspension and the optimized anti-roll bar suspension, the ride comfort of the optimized HIIRS can be improved by 22.58% and 18.41%, respectively. The road holding of the optimized HIIRS can be increased by 19.46% and 17.91%, respectively.

### 5.2. Multi-Objective Optimization

In the previous section, the single-objective optimization of ride comfort and road holding has been achieved. When more attention is paid to ride comfort or road holding and there is no strict demand for other performance, the optimal parameter configuration can be obtained through single objective optimization. When higher energy-harvesting power is expected and the demand for other performances is not so urgent, the contradiction between the three performances of the HIIRS system needs to be considered. This section would conduct multi-objective optimization with the three performances, the ride comfort, the road holding, and the energy-harvesting characteristics.

The NSGA-II algorithm is employed to solve a Pareto front for the three performances of the HIIRS. The NSGA-II algorithm was proposed by Srinivas and Deb on the basis of NSGA and proved to be superior to the NSGA algorithm by using a fast non-dominated sorting algorithm and reducing the computational complexity [34].

The Pareto front and its 2D projection are obtained and shown in Figures 9 and 10, respectively.

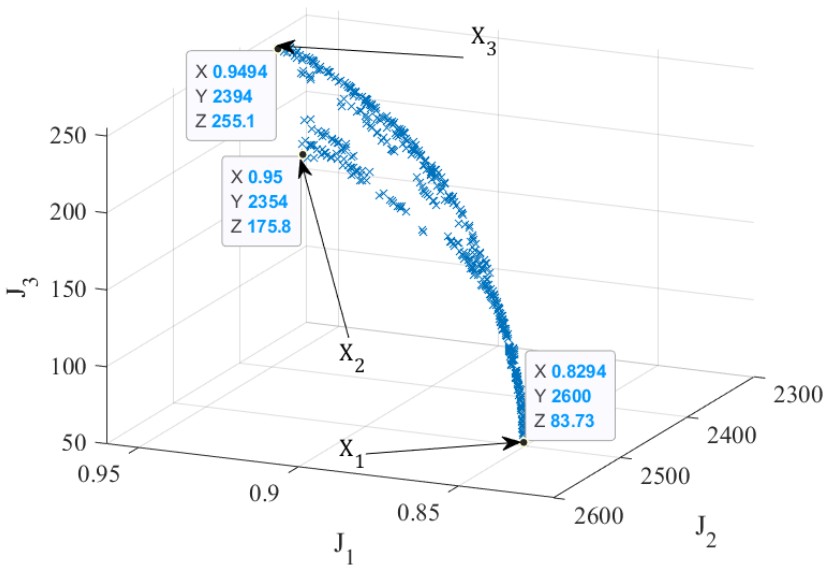

**Figure 9.** Pareto optimal set in the objective function space.

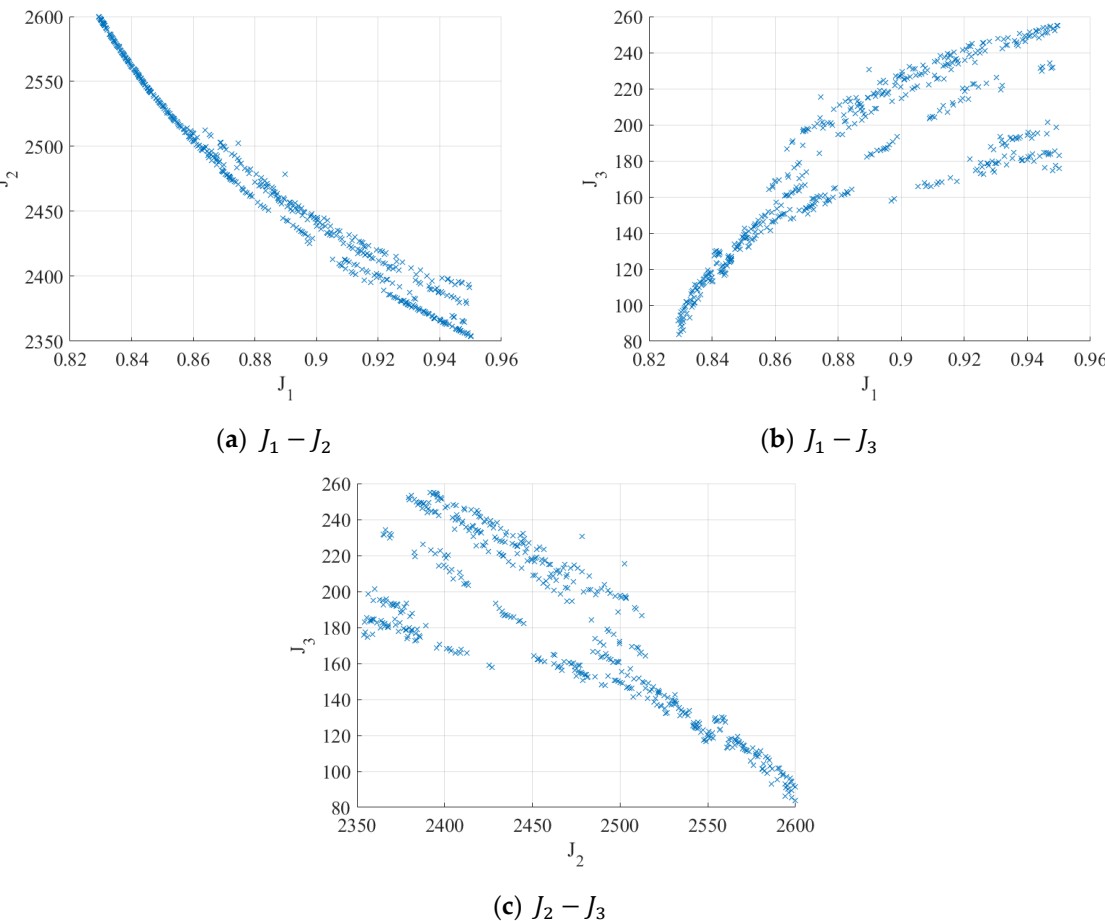

(**a**) $J_1 - J_2$

(**b**) $J_1 - J_3$

(**c**) $J_2 - J_3$

**Figure 10.** The 2D projection of the Pareto front (**a**) Pareto front of ride comfort and road holding; (**b**) Pareto front of ride comfort and energy-harvesting power; (**c**) Pareto front of road holding and energy-harvesting power.

Figures 9 and 10 show the trade-off between the three performances. Figure 10a shows that ride comfort and road holding are a pair of conflicting performance indicators. Figure 10b shows that ride comfort and energy-harvesting power also conflict with each

other. The curve in Figure 10c shows that the road holding and energy-harvesting power do not completely conflict. The three points $X_1$, $X_2$ and $X_3$ in Figure 9, respectively, represents the three parameter sets of the HIIRS with the best ride comfort, road holding, and energy-harvesting power. The three solutions and the corresponding objective function values are shown in Table 8. The PSD of acceleration and tire dynamic load and energy-harvesting power of $X_1$, $X_2$ and $X_3$ are shown in Figure 11.

**Table 8.** Optimization results.

| Parameter | Optimization Value of $X_1$ | Optimization Value of $X_2$ | Optimization Value of $X_3$ |
|---|---|---|---|
| Resistance $R$ | 4.5208 | 3.86 | 5.1703 |
| Inner diameter of hydraulic cylinder $D_c$ | 0.039 | 0.0412 | 0.0388 |
| Displacement of hydraulic motor $q_m$ | $1 \times 10^{-4}$ | $1 \times 10^{-4}$ | $2.73 \times 10^{-5}$ |
| Bounce acceleration RMS | 0.7757 | 0.9050 | 0.9050 |
| Roll acceleration RMS | 0.4699 | 0.4769 | 0.4769 |
| Total acceleration RMS | 0.8294 | 0.95 | 0.9494 |
| Tire dynamic load RMS | 2600 | 2353.7 | 2393.6 |
| Energy-harvesting power $P_t$ | 83.73 | 175.8 | 255.1 |

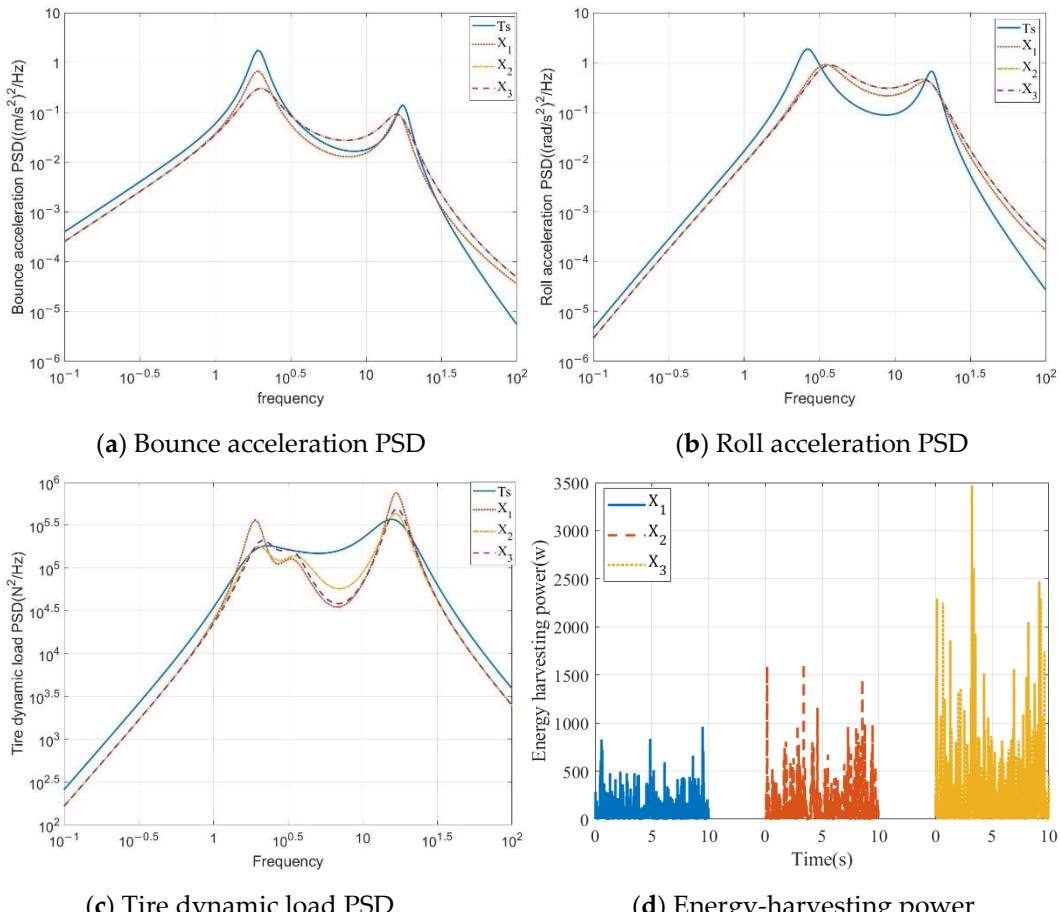

(**a**) Bounce acceleration PSD

(**b**) Roll acceleration PSD

(**c**) Tire dynamic load PSD

(**d**) Energy-harvesting power

**Figure 11.** Acceleration and tire dynamic load PSD and energy-harvesting power. (**a**) Bounce acceleration PSD; (**b**) Roll acceleration PSD; (**c**) Tire dynamic load PSD; (**d**) Energy-harvesting power.

Figure 11a illustrates the bounce acceleration of the optimized traditional suspension and the optimized HIIRS. The largest amplitude of the first peak is the traditional suspension, followed by $X_1$, and the lowest is $X_2$ and $X_3$. Compared with the parameters of $X_2$, $X_3$ has larger resistance, smaller cylinder diameter, and smaller displacement of hydraulic motor. Therefore, the bounce modes of $X_2$ and $X_3$ have the same stiffness

and damping. $X_1$ has greater resistance and a smaller hydraulic cylinder diameter than $X_2$, which makes $X_1$ have less damping. Therefore, the first peak value of $X_1$ is greater than that of $X_2$ and $X_3$.

Figure 11b shows the roll acceleration PSD. The first peak of the PSD curve of the roll acceleration of the optimized HIIRS appears around 3.1 Hz, while the first peak of the traditional suspension appears around 2.6 Hz. This is because the high-pressure oil from both hydraulic cylinders of the HIIRS is collected at the high-pressure accumulator, which provides additional roll stiffness [32].

Figure 11c shows the PSD curve of tire dynamic load. Compared with traditional suspension, the optimized HIIRS has additional peaks. The additional peak frequency occurs at 3.2–3.6 Hz. This peak frequency is close to one of the peaks in the frequency response of the displacement of the HIIRS wheels to the ground excitation on the opposite side. Therefore, it is speculated that the large displacement of the wheels under the 3.2–3.6 Hz ground excitation on the opposite side leads to the peak of the dynamic load of the wheels.

Figure 11d showed the energy-harvesting power. The resistance of $X_1$ is larger than that of $X_2$. Therefore, the energy-harvesting power of $X_1$ is less than that of $X_2$. Compared with $X_2$, the resistance of $X_3$ is higher, the cylinder diameter of $X_3$ and displacement of $X_3$ are smaller and the energy-harvesting power of $X_3$ is higher. The riding comfort and road holding of $X_2$ and $X_3$ are similar, indicating that their damping is very close. When the damping is close, it could be considered that the speed of the piston rod is close too, in this case, Equation (27) shows that the energy harvesting power is inversely proportional to the displacement and resistance and is positively correlated to the inner diameter of the hydraulic cylinder. The motor displacement of $X_3$ is much smaller than that of $X_2$ and the resistance is slightly higher, and the inner diameter is slightly smaller, the positive effect of greatly reduced displacement on the increase of energy-harvesting power exceeds the negative effect that slightly increases resistance and slightly reduces cylinder diameter caused, so the energy-harvesting power of $X_3$ is higher than that of $X_2$.

To sum up, the multi-objective optimization obtained the Pareto solution set of three performance indexes: ride comfort, road holding, and energy-harvesting power, which shows that ride comfort gets conflicted with road holding and energy-harvesting power, while road holding and energy-harvesting power does not conflict. The maximum energy-harvesting power can reach 255.1 W, and the ride comfort and road holding are improved by 7.35% and 11.25%, respectively, compared with the optimized traditional suspensions.

## 6. Conclusions

The main contribution of this paper is to provide a new idea about performance optimization for the HIIRS. With the established half-vehicle model equipped with the HIIRS, the parameter sensitivity was analyzed and the key parameters were determined for the three performance indices, the weighted vehicle body acceleration representing the ride comfort, the dynamic tire load representing the road holding, and the average energy-harvesting power. The obtained three key parameters, the inner diameter of the hydraulic cylinder $D_c$, the electric resistance in the energy-harvesting circuit $R$, and the displacement of the hydraulic motor $q_m$, were selected from the initial nine parameters with the Morris method, which greatly reduced the computational effort in the optimization. Single-objective optimization results demonstrated that the ride comfort and the road holding of the optimized HIIRS were superior to other optimized suspensions. Since the ride comfort, the road holding, and the energy-harvesting power were contradictory, a multi-objective optimization was also conducted. A Pareto front was obtained with the NSGA-II algorithm, the ride comfort conflicts with the road holding and the energy-harvesting power, while road holding and energy-harvesting power did not conflict. The Pareto and optimization results can guide the future design of the HIIRS system.

**Author Contributions:** Conceptualization, S.G. and G.T.; methodology, S.G. and L.C.; software, L.C. and X.W.; validation, S.G. and Y.P.; formal analysis, S.G., L.C. and Y.P.; investigation, S.G.

and L.C.; resources, L.C.; data curation, L.C.; writing—original draft preparation, S.G. and L.C.; writing—review and editing, S.G. and Y.P.; visualization, L.C.; supervision, S.G. and G.T.; project administration, S.G. and G.T.; funding acquisition, S.G. All authors have read and agreed to the published version of the manuscript.

**Funding:** This research was funded by the National Natural Science Foundation of China, grant number 51905394, and the Scientific Research Foundation of Wuhan University of Technology, grant number 3120622846.

**Data Availability Statement:** No new data were created or analyzed in this study. Data sharing is not applicable to this article.

**Acknowledgments:** The authors gratefully acknowledge the Hubei Key Laboratory of Advanced Technology for Automotive Components, Hubei Collaborative Innovation Centre for Automotive Components Technology.

**Conflicts of Interest:** No potential conflict of interest was reported by the authors.

## Appendix A

The description of terms in the manuscript is shown in Table A1.

**Table A1.** Descriptions for symbols in manuscript.

| Symbol | Description | Symbol | Description |
|---|---|---|---|
| HIIRS | Hydraulic integrated interconnected regenerative suspension | HIS | Hydraulic interconnected suspension |
| Ts | Traditional suspension | ARB | Anti-roll bar |
| $M$ | Sprung mass | $P_{ph}$ | Pre-charge pressure of high-pressure accumulator |
| $m_j$ | Unsprung mass ($j = l$, $r =$ left, right) | $V_{ph}$ | Pre-charge gas volume of high-pressure accumulator |
| $I$ | Sprung mass moment of inertia about the roll axis | $P_{pl}$ | Pre-charge pressure of low-pressure accumulator |
| $b_j$ | Distance from c.g. to suspension strut ($j = l$, $r$ = left, right) | $V_{pl}$ | Pre-charge gas volume of low-pressure accumulator |
| $k_{sj}$ | Mechanical suspension spring stiffness | $R$ | Circuit external resistance |
| $k_{tj}$ | Tire stiffness | $D_c$ | Inner diameter of hydraulic cylinder |
| $c_{tj}$ | Tire damping coefficient | $q_m$ | Hydraulic motor displacement |
| $C_e$ | Traditional suspension damping coefficient | $D_p$ | Inner diameter of hydraulic pipeline |
| rms | Root-mean-square | $P$ | Initial pressure of hydraulic system |

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
