# Peer review of "Hydraulic Integrated Interconnected Regenerative Suspension: Sensitivity Analysis and Parameter Optimization"

_electronics, doi:10.3390/electronics12040891_

Round 1

Reviewer 1 Report

This work performs sensitivity analysis and optimization of parameters of a recently proposed hydraulic integrated interconnected regenerative suspension (HIIRS), where ride comfort, road holding, and energy harvest are mainly concerned. Overall, this paper is well written with results clearly presented, the following comments need to be addressed or considered before publication.

1) In terms of the parameter optimization for ride comfort, are there any weighting functions included to take into account the frequency range of human perception? For example, the human is more sensitive to 4-8 Hz vertical vibration according to ISO.

2) A linearized half vehicle model has been employed in this work to evaluate performance and optimize parameters, however, are these results accurate enough as compared to the actual nonlinear model of full vehicle, where uncertainties in damping, the variation in pitch angle, etc. are presented?

Author Response

Response to Reviewer 1 Comments

Point 1: In terms of the parameter optimization for ride comfort, are there any weighting functions included to take into account the frequency range of human perception? For example, the human is more sensitive to 4-8 Hz vertical vibration according to ISO.

Response 1: Thank you for handling the review of our submission. We have considered the difference of human body's perception of acceleration at different frequencies. The most sensitive frequency range of vertical acceleration on the seat support surface is 4~12.5Hz. In the range of 4~8Hz, the human body produces resonance, and the vibration in the range of 8~12.5Hz has a great impact on the human spine. Therefore, the frequency weighting function of the vertical acceleration is determined as:

, which is mentioned in Section 3.2 in the article.

Point 2: A linearized half vehicle model has been employed in this work to evaluate performance and optimize parameters, however, are these results accurate enough as compared to the actual nonlinear model of full vehicle, where uncertainties in damping, the variation in pitch angle, etc. are presented.

Response 2: Thank you for handling the review of our submission. There are differences between the linearized frequency domain model and the nonlinear model, but because the frequency domain analysis method is a commonly used method in suspension dynamics, the research of similar methods and the reliability of the results of this method have been proved by experiments. We still use a linearized model to analyze, which can greatly improve efficiency.

Reviewer 2 Report

The paper proposes a new idea about the performance optimization of HIIRS.

Few questions are raised when determining the most important performance indices. The paper concludes that the inner diameter of the hydraulic cylinder, the capacity of the hydraulic motor and the resistance in the electrical circuit are the most sensitive parameters. 

How are the remaining parameters such accumulator precharge pressures and volumes sized? Are they sized/determined after the most sensitive parameters are set at the optimal values?

The paper also proposes a new way of optimizing the HIIRS by using Genetic Algorithm on multiple objective functions for optimization.

Determining the best solution from the three Pareto optimal set solutions needs to be elaborated for better understanding of the readers.

Overall, the paper is well written and studies an important optimization problem of HIIRS which results in improved performance of the suspension.

Author Response

Response to Reviewer 2 Comments

Point 1: How are the remaining parameters such accumulator precharge pressures and volumes sized? Are they sized/determined after the most sensitive parameters are set at the optimal values?

Response 1: Thank you for handling the review of our submission. The parameters of the accumulator are set by the common accumulator in the market. In this study, the optimal parameters of the accumulator are not studied, because the main focus of this study is on the straight driving condition, and the model used is only applicable to the straight driving condition. According to the relevant research results of the interconnected suspension, the parameters of the accumulator have a very important impact on the roll of the vehicle when turning. The accumulator parameters are not optimized after other parameters are optimized, which is also one of the follow-up objectives of our team.

Reviewer 3 Report

This paper is well written and provided detailed description of the method developed for the HIIRS model, although I do have some comments below:

1. Seems like the brackets for references are missing, which could cause some confusion for the readers.

2. Figure 3 can use a larger font size for the variables for better visualization.

3. There are a lot of symbols used for the parameters in this paper, maybe adding a list/table for the nomenclatures will help some readers quickly locate and understand them.

Author Response

Response to Reviewer 3 Comments

Point 1: 1. Seems like the brackets for references are missing, which could cause some confusion for the readers.

Response 1: Thank you for handling the review of our submission. We highlight the changes in the revised manuscript.

Point 2: Figure 3 can use a larger font size for the variables for better visualization.

Response 2: Thank you for handling the review of our submission. We highlight the changes in the revised manuscript.

Point 3: There are a lot of symbols used for the parameters in this paper, maybe adding a list/table for the nomenclatures will help some readers quickly locate and understand them.

Response 3: Thank you for handling the review of our submission. We highlight the changes in the revised manuscript. We have added TableA1 in the appendix to explain the nomenclatures in the manuscript.

Reviewer 4 Report

In this paper, the parameter sensitivity of the novel suspension system - hydraulic integrated interconnected regenerative suspension (HIIRS) system is analysed with three objectives, the ride comfort, the road holding, and the average energy harvesting power.

The paper topic is interesting, and the manuscript is comprehensive. However, the novelty of the paper is difficult to appreciate. The authors specify in the introduction that: 'In previous studies, sensitivity analysis and parameter optimization have played a huge role in suspension studies to pursue favorable performances. However, few studies about parameter analysis and optimization have been conducted for the newly proposed HIIRS.' The contribution of the presented study should be clearly specified in relation to these studies to which they refer.

Also, some issues should be reviewed before publication:

1. The abstract should be revised. Too many details are presented regarding the obtained results that do not have a place in this section of the paper, being also presented in conclusions. Rather, the novelty of the paper should be highlighted here.

2. The references should be placed in square brackets.

3. The conclusions should be presented in a concise form. Based on the results obtained some fundamental conclusions need to be drawn.

4. The text and English language in the paper must checked, edited, and corrected.

Author Response

Response to Reviewer 4 Comments

Point 1: The abstract should be revised. Too many details are presented regarding the obtained results that do not have a place in this section of the paper, being also presented in conclusions. Rather, the novelty of the paper should be highlighted here.

Response 1: Thank you for handling the review of our submission. We highlight the changes in the revised manuscript. We have revised the abstract in the manuscript as follows:

Abstract: Hydraulic integrated interconnected regenerative suspension (HIIRS) is a novel suspension system that can simultaneously harvest the vibration energy in the suspension and enhance the vehicle dynamics. The parameter sensitivity of the HIIRS system is analysed and the significant parameters are optimized in this paper. Specifically, a half-vehicle model with the HIIRS is established. Based on the model, the parameter sensitivity of the hydraulic system is analysed with three objectives, the ride comfort, the road holding, and the average energy harvesting power. The parameters considered in this study are more abundant than those in previous related studies, including hydraulic cylinder inner diameter, hydraulic motor displacement, resistance, initial system pressure and accumulator parameters. It turns out that the most sensitive parameters are the inner diameter of the hydraulic cylinder, the resistance, and the displacement of the hydraulic motor. To further study the performances that the HIIRS could present, both the single objective optimization and the multi-objective optimization problems are solved and compared with the optimized traditional suspensions. The optimized HIIRS performs better in ride comfort and road holding than the optimized traditional suspension and anti-roll bar suspension. Different from the previous suspension optimization design, multi-objective optimization not only considers the traditional performance of the suspension, but also incorporates the energy harvesting characteristics into the optimization objective. In the multi-objective optimization, a Pareto front is obtained, which shows that the ride comfort conflicts with the road holding and the energy harvesting power, while road holding and energy harvesting power did not conflict. The Pareto front shows that the optimized HIIRS is superior to the traditional suspension in ride comfort and road holding, and can also harvest considerable energy.

Point 2: The references should be placed in square brackets.

Response 2: Thank you for handling the review of our submission. We highlight the changes in the revised manuscript.

Point 3: The conclusions should be presented in a concise form. Based on the results obtained some fundamental conclusions need to be drawn.

Response 3: Thank you for handling the review of our submission. We highlight the changes in the revised manuscript. We have revised the conclusions in the manuscript as follows:

The main contribution of this paper is to provide a new idea about the performance optimization for the HIIRS. With the established half-vehicle model equipped with the HIIRS, the parameter sensitivity was analyzed and the key parameters were determined for the three performance indices, the weighted vehicle body acceleration representing the ride comfort, the dynamic tire load representing the road holding, and the average energy harvesting power. The obtained three key parameters, the inner diameter of the hydraulic cylinder , the electric resistance in the energy harvesting circuit , and the displacement of the hydraulic motor , were selected from the initial nine parameters with Morris method, which greatly reduced the computational effort in the optimization. Single-objective optimization results turned out that the ride comfort and the road holding of the optimized HIIRS were superior to other optimized suspensions. Since the ride comfort, the road holding, and the energy harvesting power were contradictory, a multi-objective optimization was also conducted. A Pareto front was obtained with the NSGA-II algorithm, the ride comfort conflicts with the road holding and the energy harvesting power, while road holding and energy harvesting power did not conflict. The pareto and optimization results can guide the future design of the HIIRS system.

Point 4: The text and English language in the paper must checked, edited, and corrected.

Response 3: Thank you for handling the review of our submission. We highlight the changes in the revised manuscript.

Round 2

Reviewer 4 Report

The authors responded appropriately to all points in the previous review. The manuscript has been much improved and can now be published.